# Assessment of Pumpkin Seed Oil Adulteration Supported by Multivariate Analysis: Comparison of GC-MS, Colourimetry and NIR Spectroscopy Data

**DOI:** 10.3390/foods11060835

**Published:** 2022-03-14

**Authors:** Sandra Balbino, Dragutin Vincek, Iva Trtanj, Dunja Egređija, Jasenka Gajdoš-Kljusurić, Klara Kraljić, Marko Obranović, Dubravka Škevin

**Affiliations:** 1Faculty of Food Technology and Biotechnology, University of Zagreb, Pierottijeva 6, 10000 Zagreb, Croatia; jgajdos2@pbf.hr (J.G.-K.); kkraljic@pbf.hr (K.K.); mobran@pbf.hr (M.O.); dskevin@pbf.hr (D.Š.); 2Department of Agriculture, Varazdin County, Franjevački trg 7, 42000 Varaždin, Croatia; dragutin.vincek@vzz.hr; 3Podravka Inc., Ante Starčevića 32, 48000 Koprivnica, Croatia; iva.trtanj@podravka.hr; 4Ledo plus Ltd., Marijana Čavića 9, 10000 Zagreb, Croatia; dunja.egredija@ledo.hr

**Keywords:** pumpkin seed oil, adulteration, NIR, colourimetry, OPLS

## Abstract

Because of its high market value, pumpkin seed oil is occasionally adulterated by cheaper refined oils, usually sunflower oil. The standard method for detecting its authenticity is based on expensive and laborious determination of the sterol composition. Therefore, the objective of this study was to determine the sterol content and authenticity of retail oils labelled as pumpkin seed oil and also to investigate the potential of near-infrared spectroscopy (NIR) and colourimetry in detecting adulteration. The results show that due to the significant decrease in Δ7-sterols and increase in Δ5-sterols, 48% of the analysed oils can be declared as adulterated blends of pumpkin seed and sunflower oil. Significant differences in NIR spectroscopy data, in the range of 904–922 nm and 1675–1699 nm, and colourimetric data were found between the control pumpkin seed oil and sunflower oil, but only the NIR method had the potential to detect the authenticity of pumpkin seed oil, which was confirmed by principal component analysis. Orthogonal projection on latent structures (OPLS) discriminant analysis, resulted in working classification models that were able to discriminate pure and adulterated oil. OPLS models based on NIR spectra were also able to successfully predict the content of β-sitosterol and Δ7,22-stigmastadienol in the analysed oils.

## 1. Introduction

In the wake of recent dietary trends that emphasise minimally processed functional foods, pumpkin seed oil is becoming one of the leading oils in this market niche. Due to the excellent gastronomic and nutritional properties of pumpkin (*Cucurbita pepo* L.) seed oil [1], the global market was valued at approximately 676.58 million USD in 2018, with an estimated growth of 15% from 2019 to 2026 [2]. The positive image that pumpkin seed oil enjoys among consumers is due to its various bioactive compounds with specific vitamin, antioxidant and pharmaceutical activity [3,4]. Pumpkin seed oil is also well known for its distinctive colour, usually described as dark green with strong red fluorescence, which is due to its dichromatic character [1]. In the production of pumpkin seed oil, ground pumpkin seeds are mixed with water and salt to form a dough which is then roasted at 110–150 °C and pressed [1]. Due to the low yield of oil pumpkin per hectare, the low yield of seed oil and the high energy requirements and the number of man-hours required for its production, pumpkin seed oil has a high market price [5]. Therefore, similar to other high value-added products, such as olive oil, honey and wine, pumpkin seed oil can be adulterated during the production process. Adulteration is most commonly done by blending with cheaper refined oils, mostly sunflower and rapeseed oils [6]. In some oil mills, the addition of other vegetable oils has even been traditionally used to facilitate roasting, i.e., to prevent the pumpkin seed dough adhesion to the pan during roasting, adding about 3 L of refined vegetable oil to 10 kg of pumpkin seeds. However, regardless of the reasons, any undeclared addition of another oil variety to pumpkin seed oil is considered fraud [7]. Nowadays, as more and more farmers and entrepreneurs choose to grow oil pumpkin and produce pumpkin seed oil to meet increasing consumer demand, the means of its authentication are also gaining importance. Austrian and Slovenian pumpkin seed oils carry the prestigious European Union quality mark assigned for Protected Geographical Indication (PGI) [8,9], while Croatian Varazdin pumpkin seed oil is protected at the national level and is currently in the process of being protected on the European level, which requires effective fraud prevention and label protection methods.

Common analytical parameters that have been proposed by various authors as markers to verify the authenticity of vegetable oils are fatty acids and triglycerides, as well as unsaponifiable compounds, such as sterols, tocopherols and tocotrienols and aliphatic alcohols [10]. However, a fundamental problem in evaluating the authenticity of a particular oil is the establishment of one or more parameters to verify its identity and purity. Ideally, the identifying characteristics should be absent in the pure and present in the adulterated oil, but this is often not the case. Therefore, it is common practice that complete profiles of certain chemical groups in suspect oil samples are determined and compared with the limits established for authentic products [11].

Due to their specific content and composition, sterols are often referred to as the “fingerprint” of edible oils, i.e., their specific distribution is characteristic for the oil type [12]. Nowadays, gas chromatographic (GC) methods for the determination of sterols are sensitive enough to indicate the addition of another oil even at low concentrations, which is why they are considered the gold standard for the determination of oil adulteration [13]. Their limits are also established as identification parameters for a variety of vegetable oils in the Codex Alimentarius food standard [11].

The sterol composition of pumpkin seed oil is very specific due to its high content of Δ7-sterols (spinasterol, Δ7-stigmastadienol, Δ7-stigmastatrienol), while refined oils potentially used as its adulterants, such as sunflower and rapeseed oils, have significantly lower levels of these sterols. Elevated levels of campesterol and β-sitosterol in pumpkin seed oil indicate adulteration, as does the presence of other phytosterols, such as brassicasterol (from rapeseed oil) or Δ5-stigmasterol, which do not occur naturally in pumpkin seed oil [6,10,14]. Since the standard method for the determination of sterols [15] is time-consuming and laborious and also often requires the use of expensive chromatographic methods, new techniques for the authentication of oils are currently being developed. These techniques include vibrational spectroscopy methods, i.e., near-infrared (NIR), Fourier transform infrared (FT-IR), and Raman spectroscopy, which can be applied to verify the authenticity of edible oils and are increasingly used due to their simplicity, speed and ease of sample preparation, are increasingly in use [16]. Among the above methods, the NIR method is highlighted because of its efficiency and is, therefore, widely used in the agri-food sector for qualitative and quantitative analyses related to food safety and quality [17]. Moreover, due to its specific needs, this sector requires the development of portable devices that offer fast, simple and non-destructive measurements that can be used in the field, in warehouses or in food quality control (control of changes in the content of certain components, freshness, adulteration, etc.) [18]. There are, however, only a few studies investigating the efficiency of rapid methods in the authenticity testing of pumpkin seed oil. These involve characterisation of fatty acid profile and δ13C isotopic ratio [19], triacylglycerols stereospecific analysis [20] and FT-IR [21]. 

The aim of this study was to (i) determine specific sterol compounds as adulteration markers in samples declared as pumpkin seed oil and collected from the Croatian market and to (ii) assess their authenticity based on the percentage of individual sterols. Sterol “fingerprints” were determined by gas chromatography coupled with mass spectrometry (GC-MS) of isolated pumpkin seed oil non-glyceride fractions. The same chemical analyses were performed on laboratory-produced pumpkin seed oil and refined sunflower oil to characterise them, i.e., to fully evaluate the properties of pure pumpkin seed oil and possible blends. In addition, due to the extreme complexity and cost of the sterol determination, the spectrum in the NIR region and colourimetric parameters in the CIE-L * a * b * system was determined in order to evaluate the potential of these rapid methods for the detection of pumpkin seed oil adulteration and also to establish their correlation with the chemical adulteration markers.

## 2. Materials and Methods

### 2.1. Oil samples

For the purpose of this research, 25 oil samples labelled as “pumpkin seed oil” or “virgin pumpkin seed oil” were collected from retail stores. The labels contained no PGI mark but also no indication of the addition of other substances, such as sunflower or other refined oils. The oils were named in the order in which they were obtained, from OS-1 to OS-25, and stored at −18 °C until analysis. In addition, a control pumpkin seed oil (PSO) was prepared in the laboratory from pumpkin seeds obtained from a local producer. A small-scale laboratory setting rather than a commercial producer was chosen for the preparation of the control sample in order to ensure the application of controlled processing conditions and to prevent accidental contamination in the vessels or pipelines. The control sample was produced according to the protocol and processing conditions commonly used in industrial production. Pumpkin seeds were ground and mixed with 20% warm water and 2% salt. The prepared dough was roasted in a laboratory conditioner set at 170 °C for 45 min, reaching a temperature of 130 °C, and pressed on a laboratory screw press (Monforts & Reiners, Lenzing, Austria). The clarified oil was then stored until analysis. Refined sunflower oil (SO) was also analysed to better characterise its properties and assess the possible adulteration of commercial oil samples.

### 2.2. Determination of Sterols

Sterols were determined according to the standard method ISO 12228-1 (2014). The method is based on saponification of the oil sample spiked with the internal standard (α-cholestanol was used instead of betulin) with ethanolic KOH solution (c = 0.5 mol L^−1^). The unsaponifiable fraction is then eluted with diethyl ether over a glass column filled with alumina. The alumina serves to retain the fatty acid anions while allowing the non-glyceride compounds, including sterols, to elute. The eluate is evaporated to dryness in a rotary vacuum evaporator at 45 °C. Sterol fraction, containing Δ5- and Δ7-sterols, is then isolated by thin-layer chromatography on a silica gel chromatography plate using a hexane and diethyl ether (1:1, *v*/*v*) solvent system. The sterol zone is scraped off the developed silica plate and rinsed with diethyl ether through a filter. The solvent is again evaporated with a rotary evaporator and purged to dryness under a stream of nitrogen. The sterol fraction is silylated with a mixture of pyridine, hexamethyldisilazane and trimethylchlorosilane at a ratio of 5:2:1 (*v*/*v*/*v*), heated at 105 °C for 15 min, cooled and centrifuged. The supernatant was analysed using Agilent Technologies 6890 N Network GC System (Agilent, Santa Clara, USA) gas chromatograph equipped with DB-17MS (Agilent, Santa Clara, USA), 30 m × 0.32 mm, 0.25 μm (50%-phenyl)-methylpolysiloxane and flame ionisation detector (FID). Heating at a rate of 6 °C/min was applied from 180 to 270 °C, where it was kept for 30 min. Helium was used as the carrier gas at a flow rate of 1.5 mL/min, while the injector temperature was set at 290 °C, the split ratio at 13.3:1 and the detector at 280 °C. Quantification was performed via the internal standard method. The 5973 mass detector (Agilent, Santa Clara, USA) with the transfer line maintained at 280 °C, the MS source at 230 °C and the quadrupole at 150 °C, was used for identification purposes. Mass spectra from which sterols were identified using the NIST 17 data library were scanned in the range 30–550 (m/z).

### 2.3. Colourimetric Determinations

A Minolta CR-400 colourimeter (Minolta Camera Co. Ltd., Osaka Japan) was used to perform the analysis. Before each series of measurements, the colourimeter was calibrated on a white plate. The colour of the oil samples was measured in a glass cuvette with a volume of 5 mL and a width of 5 mm. Tristimulus system data for visual colour matching under standardized conditions were used to obtain L*, a* and b* values according to the method EN ISO 11664-4 [22]. The L* value determines whether an object is dark or light. If L* = 0, the object is black, and if L* = 100, the object is white. The a* value determines whether an object is red or green. If a* is positive, the object is red, while if a* is negative the object is green. On the other hand, the b* value determines whether an object is yellow or blue. If b* is positive, the object is yellow, and if b* is negative, the object is blue. Based on the values obtained for oil samples labelled as PSO and refined SO (L_n_, a_n_ and b_n_), the total colour difference (TCD) was calculated with respect to laboratory-produced PSO (L_PSO_, a_PSO_ and b_PSO_) as follows:(1)$$\mathrm{TCD}=\sqrt{{\left(L_{n}-L_{PSO}\right)}^{2}+{\left(a_{n}-a_{PSO}\right)}^{2}+{\left(b_{n}-b_{PSO}\right)}^{2}}$$

### 2.4. NIR Measurement

For the NIR spectroscopy of oil samples, a Control Development Inc. NIR spectrometer, IR-128-1.7-USB/6.25/50 μm was used. This is a benchtop NIR spectrometer whose main components are a light source (tungsten-halogen lamp), a NIR spectrometer and a cuvette sample holder (5 mm wide slit and a lid). The components are connected by optical cables, and the NIR instrument is connected to a computer with installed Control Development software Spec32. All measurements were performed in triplicate in the near-infrared wavelength range from 904 to 1699 nm at ambient temperature. For each sample, the average NIR scan was calculated. Each sample was presented with 4 scans (3 original and one average) for 796 data of wavelengths (used step: 1 nm).

### 2.5. Statistical Analysis

All analytical measurements were performed in triplicate and used to calculate the mean values. Principal component analysis (PCA) was performed separately for colourimetric and NIR data at the 95% significance level using XLSTAT 2019.3.2 software, which was also used to determine the correlation between individual sterols. Orthogonal projection on latent structures, also known as partial least squares regression discriminant analysis (OPLS-DA), was performed on the NIR spectroscopy data to build a classification model that would distinguish between pure and adulterated PSO based on these parameters. In addition, orthogonal projection to latent structures (OPLS) was used to explore the predictive ability of NIR spectroscopy data for the content of individual sterol compounds, i.e., to create models able to predict β-sitosterol and Δ7,22,25-stigmastatrienol content in oils. Both OPLS-DA and OPLS were performed in Simca 17.0.2. software package with cross-validation set to 7 groups.

## 3. Results and Discussion

### 3.1. Total Sterols

Figure 1 shows a Box-whisker plot of the total sterol contents in the analysed oil samples labelled as pumpkin seed oils, with a mean of 2122.61 mg/kg and a range of 1765.10–2532.00 mg/kg, while the first and third quartiles were 1995.83 and 2246.18 mg/kg, respectively, and the median was 2082.75 mg/kg. Since Codex Alimentarius [11] does not contain authenticity parameters for pumpkin seed oil, when these results are compared with the lower limit for total sterol content, which is set at 2100–5600 mg/kg of oil in the Croatian national Regulation [23], up to 15 oils can be declared non-compliant. While this result could indicate adulteration of these non-compliant samples, sunflower oil as a possible adulterant and also laboratory-produced pumpkin seed oil contained similar levels of total sterols (2250.2 and 2284.2 mg/kg, respectively) also with similar regulation [23] limits (2400–5000 mg/kg). Taking this into consideration, the observed decrease in total sterol content alone cannot be considered a reliable indicator of authenticity. The reasons for this lower total sterol content could be related to the selection of different oil pumpkin cultivars used for pumpkin seed oil production, as well as to possible changes caused by cultivation conditions associated with the crop season [24]. Similar values, lower than the limit prescribed by the Regulation [23] were also observed by Murkovic et al. [25], who found that the content of total sterols in the pumpkin seed oils analysed in their study was 1930 mg/kg, which is close to the average value obtained in this study.

### 3.2. Individual Sterols

Breinhölder et al. [26] concluded that pumpkin seed oils differ significantly in terms of Δ7-sterol content from most known vegetable oils, which contain predominately Δ5-sterols. On the other hand, Δ5-sterols are characteristic and present in high amounts in most common refined oils, such as sunflower, rapeseed and soybean. Authors Dulf et al. [27], in their study on the changes in the content of sterols when pumpkin seed oil was blended with sunflower oil, came to similar conclusions and found that the content of Δ5-sigmasterol and β-sitosterol increased when 30% cold-pressed sunflower oil was added to cold-pressed pumpkin seed oil. 

In a study by Mandl et al. [6], the authors stated that the presence of brassicasterol (in the case of the addition of rapeseed oil) or Δ5-stigmasterol (in the case of the addition of sunflower oil) may indicate adulteration of pumpkin seed oil, even with the addition of only 2% other oil. In the analysed oils, brassicasterol was not detected in any sample of commercial oils labelled as PSO, which means that they were not adulterated by the addition of rapeseed oil, but only sunflower oil.

The GC-MS analysis used to “fingerprint”, i.e., to evaluate the content and composition of the analysed oil samples was able to distinguish nine individual sterol compounds. The individual sterols determined by GC-MS in the analysed samples of commercial oils labelled as PSO, laboratory pumpkin seed oil and refined sunflower oil are divided into Δ5 (4 compounds) and Δ7 (5 compounds)-sterols and are listed in Table 1 and Table 2. From the results for the content of Δ5-sterols (Table 1), sunflower oil contains 90.18% Δ5-sterols, while laboratory PSO contains 8.71%, which is consistent with the literature [28,29], i.e., it confirms the differences in the composition of these two oils.

The predominant Δ5-sterol in the refined sunflower oil sample was β-sitosterol, with a content of 61.21%, while it was 5.66% in the laboratory-produced PSO. Other Δ5-sterols detected in both oils were campesterol, Δ5-stigmasterol and campestanol. The Croatian Regulation [23] sets the limit for the content of β-sitosterol in pumpkin seed oil at 1–8%, while campesterol and Δ5-stigmasterol are set at 1–5%, i.e., according to these proscribed parameters, the laboratory-produced PSO is recognized as authentic. 

The content of total Δ5-sterols in 25 commercial samples labelled as PSO ranged from 4.20% to 42.43%. The highest single content of β-sitosterol (30.56%), which was six times higher than the identification limit specified in the Regulation [23], was detected in the sample OS-12. Overall, large deviations in the content of total Δ5-sterols (>36%) were detected in 12 commercial oils labelled as PSO, indicating the addition of variable amounts of sunflower oil.

On the other hand, the determined contents of dominant Δ7-sterols, i.e., Δ7,25-stigmastadienol, Δ7,22,25-stigmastatrienol and spinasterol (Table 2), in the laboratory PSO sample are in agreement with the literature data [27,28], but also, as in the case of β-sitosterol, a considerable number of samples of commercial oils do not comply with certain limits of the Regulation [23]. The content of spinasterol is lower than the lower limit in 12 samples; the share of Δ7,22,25-stigmastatrienol was within the limits in laboratory PSO and another 17 commercial samples, while for the share of Δ7,25-stigmastadienol samples do not agree with the established limits.

Although this finding leads to the conclusion that a revision of the Regulation [23] is necessary, based on the Δ5 and Δ7-sterol content, it can be stated beyond doubt that 12 oils (48%) with significantly elevated levels of Δ5-sterols are blends of pumpkin and sunflower oils, i.e., that so many samples included in this study were undoubtedly adulterated. According to the results of this analysis, it can be roughly estimated that about 30–40% sunflower oil was added to pumpkin seed oil.

### 3.3. NIR Spectroscopic Analysis

The results of NIR spectroscopic analysis of the samples of laboratory PSO and SO are shown in Figure 2. From the spectra shown, which were taken at wavelengths from 904 to 1699 nm, it is apparent that they are very similar for the most part. Differences between the samples are visible at wavelengths 904–922 nm (C-H and O-H third overtones) and 1675–1699 nm (ArC-H and C-H first overtones). A comparison of the spectra of laboratory-produced PSO and refined sunflower oil shows that the absorption values at the indicated wavelengths are lower in sunflower oil than in pumpkin seed oil. The differences in the two spectral regions mentioned above can be associated with the significantly higher values of Δ5-sterols in the SO sample (except campestenol) and Δ7-sterols in the PSOs (except Δ7-stigmasterol), which can be seen in Table 1 and Table 2. Studies on papaya and seeds [30] and other edible oils [31] suggest that the wavelengths shown in Figure 2 (904–922 and 1675–1699 nm), which distinguish PSO from SO, can really be considered imprints of the origin of the seed oil, and in this case, they are the fingerprints of oils produced from pumpkin seeds. This difference makes NIR spectroscopy a potential tool for detecting pumpkin seed oil adulteration, as it has been previously shown for a number of other high-value unrefined oils [32]. The spectral data of pumpkin seed oil have hardly been investigated so far. Lankmayr et al. investigated the capabilities of UV-Vis and infrared chemometrics for the quality classification of pumpkin seed oils and proved its efficiency through linear discriminant analysis models. However, their work focused on the classification of pumpkin seed oil as “good” and “bad” based on its sensory characteristics determined by the panel of expert evaluators rather than detecting adulteration. In addition, their work included measurements at 3074 and 10,001 nm and cannot be compared to the results of the present study because the regions do not overlap [33].

On the other hand, similar to the present study, Inarejos-Garcia et al. found that virgin olive oils of different quality classes differed in the 1700 nm region, which they related to stretching vibrations of methyl, methylene and ethylene groups [34]. Comparison of NIR spectroscopy data of sunflower, maize and olive oils also revealed large differences between samples at about 1650 nm [35], as well as sesame, safflower, mustard, peanut, olive, canola and soybean oils [36]. Due to its size, the NIR instrument that was used for the present study could be considered portable. Therefore, it must be emphasised that miniaturised NIR spectrometers have their advantages and disadvantages [37]. Portable NIR instruments provide on-site analysis but are much less uniform, resulting in different performance (narrower spectral ranges, lower spectral resolution, accuracy and reliability levels) [38]. The measurement range of the instrument used, i.e., from 900 to 1700 nm, is a narrower NIR range (typically 800–2500 nm); therefore, it is also important to determine the acceptability of this measurement range in scientific research of various foods [39], as was the case with pumpkin seed oil in this study.

### 3.4. Colourimetric Analysis

The concentrations and compositions of natural pigments in fats are an important quality parameter because they affect the colour, one of the basic attributes by which consumers judge product quality (e.g., green olive oil, red palm oil, yellow-yellow cottonseed oil). The colourimetric method was applied due to the significant difference in the colour of pumpkin seed oils and refined sunflower oil, which can be observed from the analytical results presented in Figure 3. 

The brightness or L* value represents a scale from 0 to 100, ranging from black to white. It was 42.99 for laboratory-produced PSO and 46.53 for refined sunflower oil, confirming a clearly visible difference in the brightness of pumpkin and sunflower oils. The L* values in the samples of commercial oils ranged from 42.66 to 43.81. Measured values for a*, where negative values generally correspond to a green colour and positive values to red, was −0.05 for laboratory-produced PSO, while it was much higher for sunflower oil at 0.94. The reason for such a* values of PSO could be the aforementioned dichromatism that occurs in this oil [40]. a* values in commercial oil samples labelled as PSO ranged from −0.20–0.49. The b* value represents a scale that ranges from blue (negative value) to yellow (positive value), and in this study, it is 4.14 for laboratory-produced PSO and 4.22 for refined SO. The values for the other samples range from 2.28 to 5.37.

Andjelkovic et al. [41] studied the content of phenolic compounds in six samples of PSO and its correlation with certain quality parameters, including colour. The L* value of pumpkin seed oil ranged from 43.09 to 49.53 and was not significantly different, while the values of a* (−0.15–3.44) and b* (−0.32–8.47) were significantly different. In addition, the total phenolic composition was found to be negatively correlated with L*, b* and calculated saturation and positively correlated with a* and calculated hue. Their results, except for the extremely high values for L*, a* and b* in one sample, are consistent with the results of this study.

Finally, the TCD value, which represents the total colour difference compared to a reference sample, is calculated using the L*, a*, and b* values. If it is greater than 0.2, it means that the colour difference between the two objects can be perceived by the human eye. Laboratory0produced PSO was used as a reference for the comparison and calculation of TCD. As expected, the TCD value for SO was the largest, 3.65. The TCD values obtained for the samples of commercial oils labelled as PSO ranged from 0.16 to 1.96 but were around 1 for the vast majority of the samples, which means that the colour differences are only weakly perceptible to the human eye.

### 3.5. Principal Component Analysis

To preliminarily analyse the differences between samples and to reduce the data dimensionality, principal component analysis (PCA) of the determined NIR spectroscopy and colourimetric data (variables) was performed in commercial oils labelled as PSO, laboratory PSO and refined sunflower oil (cases). In relation to the sterol analysis results, commercial oils labelled as PSO were divided into two classes: (i) pure PSO with β-sitosterol values < 15% (ranging 3.31–14.39%) and (ii) adulterated PSO with β-sitosterol values > 15% (ranging 27.00–30.56%). β-sitosterol was chosen as the threshold because of its predominance in SO and also a high correlation with the other sterols. Correlation coefficients determined for β-sitosterol were r = 0.909 for campesterol, 0.706 for stigmasterol while negative correlation was observed with spinasterol (r = −0.967), Δ7,22,25-stigmastatrienol (r = −0.949) and Δ7,25-stigmastadienol (r = 0.737). The pure PSO class was, therefore, characterised by the total Δ5-sterols content ranging from 4.20% to 20.83%, while their content was 36.03–42.43% in the adulterated PSO class. The results are presented as projections of variables (Figure 4a and Figure 5a) and samples (Figure 4b and Figure 5b) in the principal components space.

For the NIR results, a total of 44 variables (wavelengths) and 27 cases (samples) were included in the PCA analysis. The PCA run yielded two principal components, i.e., factors (F1 and F2) with eigenvalues of 41.627 and 2.217, respectively, which explained 99.65% of total data variance. The first factor (F1) explained 94.61% of the total variance and was strongly positively correlated with all NIR spectroscopy data. On the other hand, F2 explained 5.04% of the variance and showed a weak positive correlation with the NIR values measured at 1675–1681 nm. Looking at the distribution of samples in the factorial plane (F1×F2), a very clear separation of the samples of laboratory-produced pumpkin seed oil (PSO) and refined sunflower oil (SO) can be seen along both factors with PSO in the fourth and SO in the second quadrant of the factorial plane (Figure 4b). Commercial oil samples labelled as PSO are also clearly divided into two groups, with the first group (green) characterised by positive F2 values and consisting of oil samples classified as pure PSO based on the analysis of sterols, and the second group (orange) containing samples for which adulteration by blending with sunflower oil has been demonstrated beyond doubt by elevated Δ^5^-sterol content. 

The PCA analysis for colourimetric data was performed with 4 variables and 27 cases. The first two factors explained 91.80% of the total variance, with F1 explaining 63.88% and F2 27.92%. F1 had a strong positive correlation with L* and TCD and a strong negative correlation with a*, whereas b* was strongly correlated with F2 (Figure 5a). In contrast to NIR, the colourimetric data could not distinguish between pure and adulterated oil samples, although the laboratory-produced PSO and SO were placed on the opposite sides of the factorial plane. Even though there is evidence that colourimetric parameters can be successfully used to detect adulteration in olive and sunflower oils [42,43], the present study shows that the colour of pumpkin seed oil cannot be used to assess its purity. Since the colour of PSO is highly dependent on the processing conditions, especially roasting time and temperature, which lead to differences in the extraction of chemical compounds from the seeds into the oil [25], there is a high variance in the colour, so the addition of much lighter SO cannot be detected through these parameters. 

### 3.6. OPLS-DA

Based on the obtained PCA results and the observed differences in NIR spectroscopy of laboratory-produced PSO and refined SO, OPLS-DA of these data was performed with the aim of producing a classification model capable of distinguishing pure and adulterated PSO based on these parameters. As proposed by Hair et al., PLS was chosen as the chemometric method able to handle small sample sizes better than the covariance-based approaches [44]. For the samples entering OPLS-DA, a class (pure or adulterated) was assigned based on the sterol’s composition. OPLS-DA yielded a model fitted with one predictive and three orthogonal components (R2X = 0.999, R2Y = 0.766, Q2 = 0.633) for which the score plot (Figure 6) shows distinction of the sample classes along the first factor, with the pure samples characterised by negative and adulterated samples by positive values. Variable importance in projection (VIP) scores for the values measured at 1675–1699 nm were >1, indicating their significance for the discrimination of the results. The S-plot of the model (data not shown) shows that adulterated blends are characterised by higher NIR spectrometry values at 1675–1699 nm. To confirm the efficiency of this discrimination method, the samples were randomly divided into a training set (20) and a test set (5). The model created proved to be suitable for class prediction (R2X = 0.999, R2Y = 0.806, Q2 = 0.735), with all tested samples correctly classified as pure or adulterated PSO, i.e., the rate of correct classification rate was 100%. These results are in agreement with recent studies of Dogruer et al. [45], who investigated the possibility of authenticating different cold-pressed oils by chemometric techniques based on FTIR and fluorescence spectroscopy. The OPLS-DA model they developed was able to detect adulteration of cold-pressed pumpkin seed oil at a content higher than 1%. The adulteration of pumpkin seed oil with sesame oil was also successfully predicted by discriminant analysis of FTIR spectra [46].

### 3.7. OPLS

Furthermore, to gain insight into the relationship between the NIR data and the results of the GC analysis of the sterols, PLS regression was performed. This statistical method also allowed the establishment of correlations between actual and predicted percentages for β-sitosterol and Δ7,22,25-stigmastatrienol (Figure 7 and Figure 8).

In the case of β-sitosterol, the PLS model showed good correlation in both calibration and cross-validation (R2 = 0.999, R2Y = 0.779, Q2 = 0.724), while its RMSEE and RMSEcv values were similar at 6.470 and 6.668, respectively. Moreover, the ratio of predicted deviation (RPD) was 2.152, which according to Torres Mariani et al. [47], falls within the range that allows approximate quantitative prediction. A good PLS model correlation was also achieved for Δ7,22,25-stigmastatrienol (R2 = 0.999, R2Y = 0.750, Q2 = 0.685) with RMSEE = 2.298, RMSEcv = 2.792 and RPD = 2.000, which is slightly lower compared to the β-sitosterol model. For both models, the VIP scores were >1 for NIR values measured at 904–922, 1698 and 1699 nm. The estimation of various chemical compounds and parameters in pumpkin seed oils was previously performed by using FTIR and fluorescence spectra [45]. The obtained models were very successful in determining fatty acids present at higher concentrations, such as palmitic, oleic and linoleic acids. On the other hand, Irnawati et al. [48] successfully used FTIR spectra to predict the antioxidant activity of pumpkin seed oils produced by pressing microwave pretreated pumpkin seeds.

## 4. Conclusions

Analysis of fingerprinting sterol compounds showed a significant increase in the content of Δ5-sterols and a decrease in the content of Δ7-sterols in 12 of 25 samples collected at retail and labelled as PSO, based on which it can be roughly estimated that these samples contain 30–40% sunflower oil. The issue of pumpkin seed oil identification is regulated by the national legislation [23]; however, the discrepancies found between the levels of individual sterols determined in the control PSO sample and set identification limits urge caution and further detailed studies that would allow the establishment of accurate markers and limits that consider the characteristics of changing climatic growing conditions and recently developed oil pumpkin cultivars. Furthermore, the NIR spectra of laboratory-pressed PSO and refined SO showed significant differences in the range of 904–922 and 1675–1699 nm related to C-H and O-H third overtones and ArC-H and C-H first overtones, respectively, providing an adequate PSO fingerprint as it has previously been shown for other vegetable oils [30,31]. This was also demonstrated by PCA of the NIR results for the samples collected from the retail, showing a clear grouping consistent with the results of authenticity assessment by sterol determination. However, since no such grouping was observed in the PCA of the colourimetric results despite the observed differences in the colourimetric parameters between the laboratory PSO and the refined SO, the chemometric analysis was continued with the NIR results only. OPLS-DA resulted in models that were suitable to correctly classify the groups of the pure and adulterated pumpkin seed oil determined by sterol analysis, with the NIR values measured at 1675–1699 nm showing the highest significance in distinguishing between the classes. The OPLS results also showed the ability to successfully predict the content of β-sitosterol and Δ7,22,25-stigmastatrienol based on the obtained NIR results. In conclusion, this work demonstrates that OPLS-DA can be successfully used to build a discriminative model with good predictive power that can accurately discriminate between pure and adulterated PSO based on the NIR spectroscopy data. Together with the results of similar research [21,33], this opens up room for further future research in this area. Further development and implementation of this method could enable rapid, cost-effective and accurate screening of pumpkin seed oil authenticity on a larger scale and greatly facilitate quality control.

## Figures and Tables

**Figure 1 foods-11-00835-f001:**
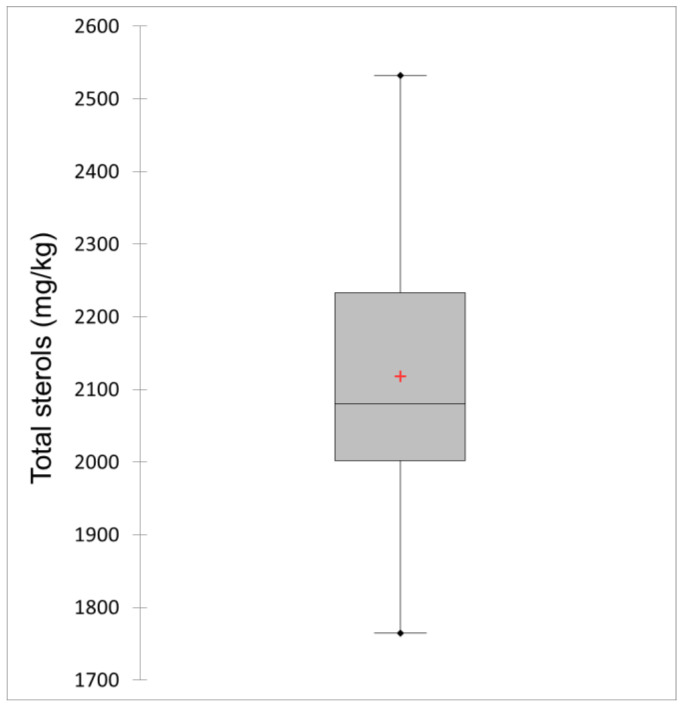
Box-whisker plot of total sterol content in the analysed oil samples labelled as pumpkin seed oils (*n* = 25).

**Figure 2 foods-11-00835-f002:**
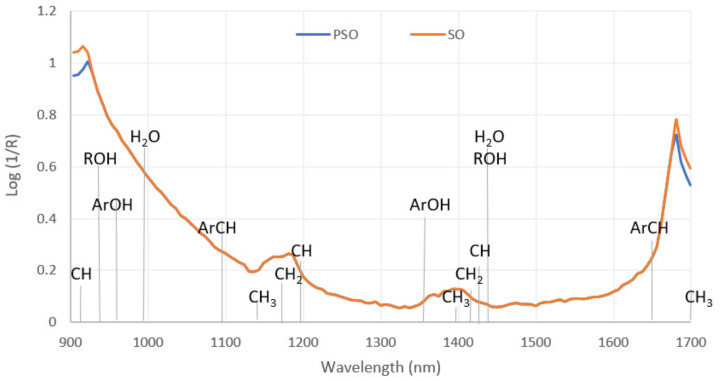
NIR spectra of samples of laboratory-pressed PSO and refined SO.

**Figure 3 foods-11-00835-f003:**
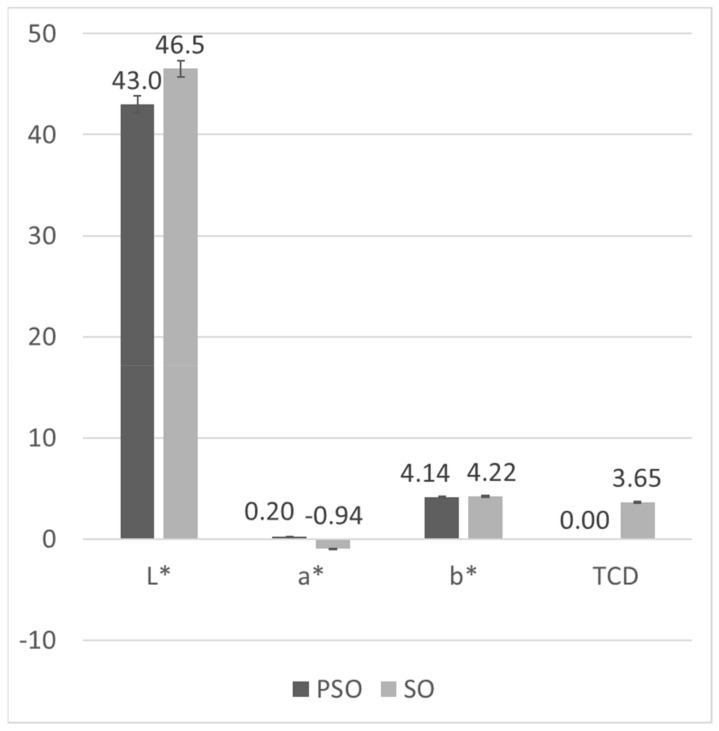
CIE colourimetric parameters of samples of laboratory-pressed PSO and refined SO (columns represent means and error bars represent standard deviation of *n* = 3 replications).

**Figure 4 foods-11-00835-f004:**
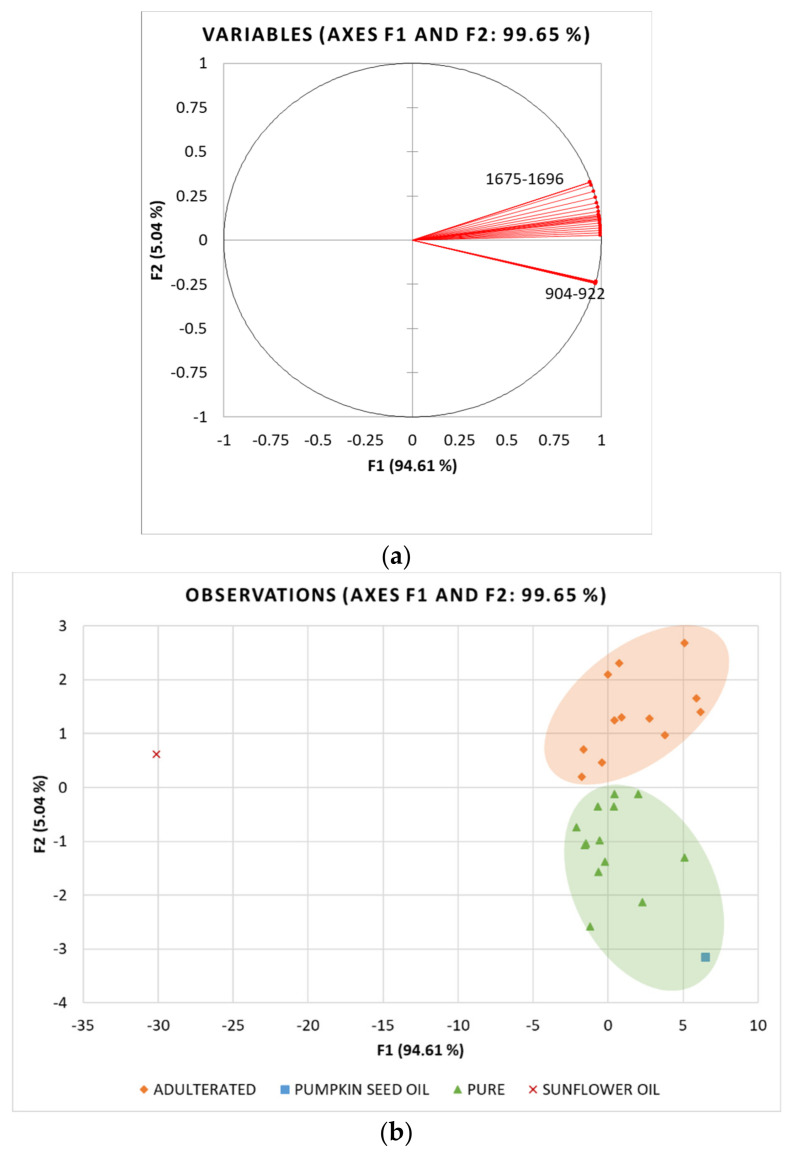
Principal component analysis (PCA) of NIR spectroscopy data: Projection of (**a**) active variables and (**b**) oil samples (SO_1–25—commercial oils labelled as pumpkin seed oil, laboratory pumpkin seed oil and refined sunflower oil) on the factorial plane (F1 × F2).

**Figure 5 foods-11-00835-f005:**
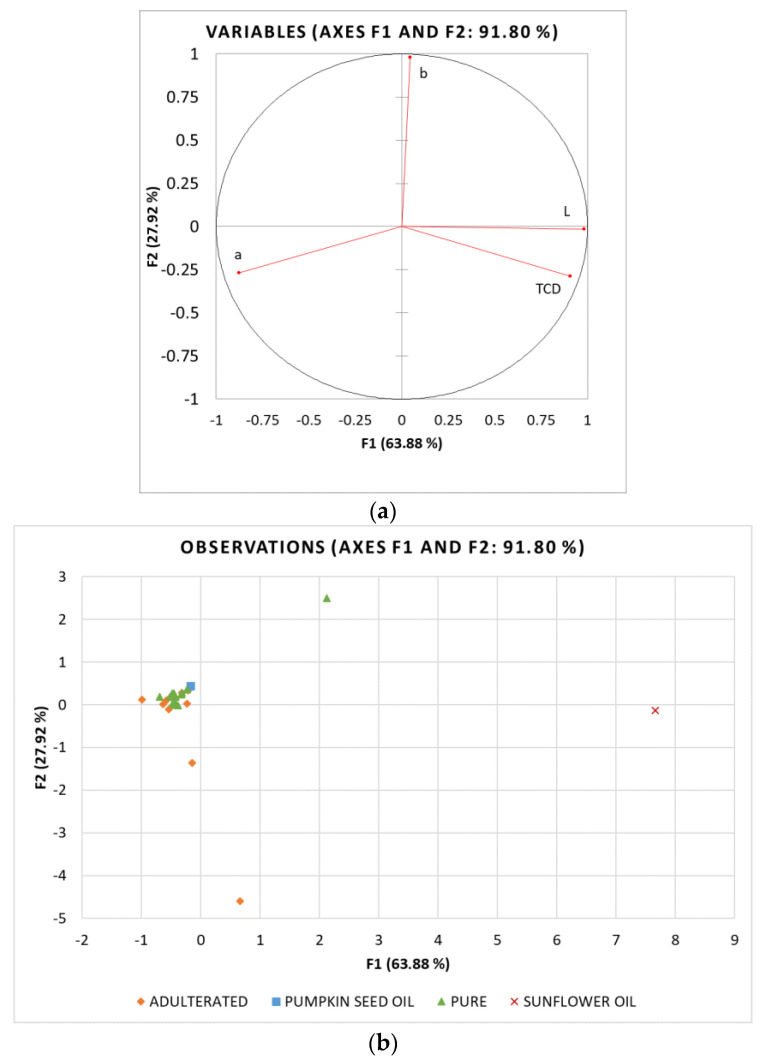
Principal component analysis (PCA) of colourimetric data: Projection of (**a**) active variables and (**b**) oil samples (SO_1–25—commercial oils labelled as pumpkin seed oil, laboratory pumpkin seed oil and refined sunflower oil) on the factorial plane (F1 × F2).

**Figure 6 foods-11-00835-f006:**
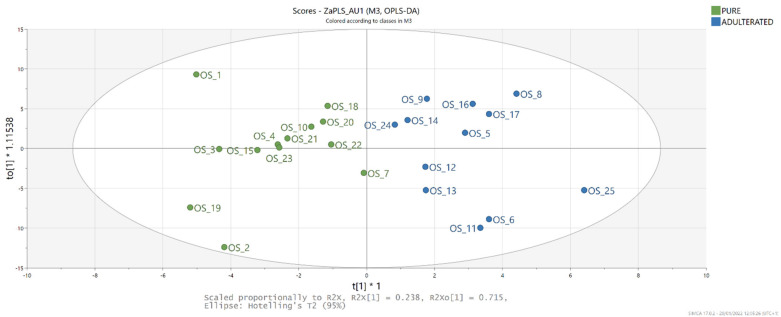
Score plot of OPLS-DA model built with NIR spectrometry data for discrimination between pure and adulterated pumpkin seed oil.

**Figure 7 foods-11-00835-f007:**
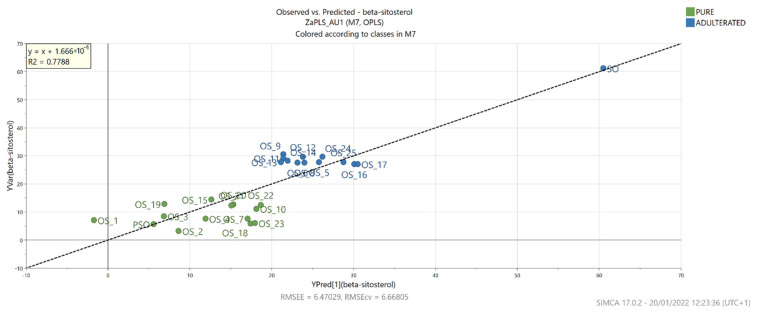
Actual and predicted β-sitosterol levels based on the OPLS model built with NIR spectrometry data.

**Figure 8 foods-11-00835-f008:**
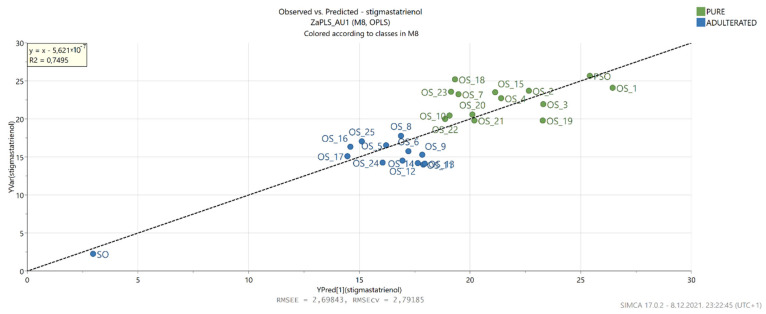
Actual and predicted Δ7,22,25-stigmastatrienol levels based on the OPLS model built with NIR spectrometry data.

**Table 1 foods-11-00835-t001:** Δ5-sterols (mean ± standard deviation, *n* = 3) in analysed oil samples (OS_1–25—samples labelled as pumpkin seed oils obtained from market, PSO—pumpkin seed oil produced in a laboratory, SO—refined sunflower oil).

Sample	Campesterol (%)	Campestanol (%)	Δ5-Stigmasterol (%)	β-Sitosterol (%)	Total Δ5-Sterols (%)
OS_1	2.23 ± 0.04	0.77 ± 0.08	2.02 ± 0.06	7.03 ± 0.12	12.05 ± 0.30
OS_2	0.00 ± 0.00	0.00 ± 0.00	0.89 ± 0.04	3.31 ± 0.16	4.20 ± 0.71
OS_3	2.78 ± 0.08	0.84 ± 0.02	1.30 ± 0.04	8.50 ± 0.11	13.42 ± 0.25
OS_4	2.77 ± 0.13	0.52 ± 0.04	3.16 ± 0.15	7.61 ± 0.27	14.06 ± 0.59
OS_5	4.75 ± 0.06	0.59 ± 0.05	5.25 ± 0.22	27.79 ± 0.24	38.37 ± 0.57
OS_6	6.27 ± 0.10	0.00 ± 0.02	5.67 ± 0.17	27.65 ± 0.49	39.59 ± 0.78
OS_7	2.05 ± 0.04	0.00 ± 0.04	0.94 ± 0.06	7.61 ± 0.09	10.61 ± 0.23
OS_8	5.92 ± 0.09	0.38 ± 0.01	5.82 ± 0.09	27.54 ± 0.27	39.67 ± 0.46
OS_9	5.70 ± 0.07	0.70 ± 0.06	5.47 ± 0.13	30.56 ± 0.33	42.43 ± 0.59
OS_10	3.02 ± 0.13	0.74 ± 0.05	1.55 ± 0.05	11.11 ± 0.41	16.42 ± 0.63
OS_11	5.17 ± 0.03	0.65 ± 0.04	5.41 ± 0.20	29.21 ± 2.56	40.44 ± 2.83
OS_12	5.06 ± 0.13	0.61 ± 0.03	0.65 ± 0.12	29.71 ± 0.18	36.03 ± 0.75
OS_13	4.30 ± 0.18	0.31 ± 0.02	5.83 ± 0.36	27.75 ± 0.30	38.20 ± 0.87
OS_14	6.11 ± 0.25	0.82 ± 0.06	5.35 ± 0.29	28.34 ± 0.20	40.62 ± 0.81
OS_15	1.18 ± 0.22	0.06 ± 0.06	2.28 ± 0.18	14.39 ± 0.68	17.92 ± 1.14
OS_16	5.03 ± 0.21	0.63 ± 0.04	4.90 ± 0.06	27.02 ± 0.13	37.59 ± 0.44
OS_17	5.03 ± 0.07	0.85 ± 0.09	5.13 ± 0.09	27.00 ± 0.74	38.02 ± 0.39
OS_18	1.92 ± 0.27	0.65 ± 0.18	1.04 ± 0.20	5.91 ± 0.41	9.52 ± 1.06
OS_19	3.20 ± 0.20	0.25 ± 0.05	2.28 ± 0.11	12.85 ± 0.41	18.58 ± 0.77
OS_20	3.74 ± 0.23	0.80 ± 0.15	3.60 ± 0.45	12.69 ± 0.49	20.83 ± 1.31
OS_21	3.15 ± 0.06	0.63 ± 0.06	2.02 ± 0.10	12.33 ± 0.44	18.13 ± 0.66
OS_22	3.58 ± 0.16	0.34 ± 0.02	2.28 ± 0.15	12.55 ± 0.33	18.75 ± 0.66
OS_23	1.83 ± 0.14	0.00 ± 0.01	0.84 ± 0.07	6.12 ± 0.27	8.79 ± 0.49
OS_24	5.35 ± 0.15	0.55 ± 0.03	0.82 ± 0.16	29.62 ± 0.48	36.33 ± 0.81
OS_25	5.18 ± 0.09	0.51 ± 0.05	4.68 ± 0.26	27.76 ± 0.33	38.14 ± 0.73
Range	0.00–6.27	0.00–0.85	0.65–5.83	3.31–30.56	4.20–42.43
PSO	1.61 ± 0.11	0.72 ± 0.08	0.72 ± 0.29	5.66 ± 0.12	8.71 ± 0.60
SO	8.54 ± 0.14	0.31 ± 0.07	10.13 ± 0.12	61.21 ± 0.02	80.19 ± 0.36

**Table 2 foods-11-00835-t002:** Δ7-sterols (mean ± standard deviation, *n* = 3) in analysed oil samples (PSO_1–25—samples labelled as pumpkin seed oils obtained from the market, PSO—pumpkin seed oil produced in a laboratory, SO—sunflower oil).

Sample	Spinasterol (%)	Δ7,22,25-Stigma- statrienol (%)	Δ7,25-Stigma- stadienol (%)	Δ7-Stigma- sterol (%)	Δ7-Avena- sterol (%)	Total Δ7-Sterols (%)
OS_1	22.01 ± 0.29	24.11 ± 0.45	25.76 ± 0.68	1.59 ± 0.15	14.48 ± 0.27	87.95 ± 2.42
OS_2	26.33 ± 0.76	23.72 ± 0.47	26.13 ± 0.57	0.97 ± 0.07	18.66 ± 0.36	95.80 ± 1.69
OS_3	23.21 ± 0.26	21.97 ± 0.30	24.92 ± 0.75	1.17 ± 0.14	15.30 ± 0.16	86.58 ± 1.61
OS_4	22.86 ± 0.28	22.76 ± 0.17	22.48 ± 0.62	1.58 ± 0.01	16.27 ± 0.31	85.94 ± 1.40
OS_5	14.38 ± 0.31	16.53 ± 0.09	16.64 ± 0.18	0.54 ± 0.04	13.54 ± 0.31	61.63 ± 0.92
OS_6	17.15 ± 0.40	15.77 ± 0.20	13.66 ± 0.33	0.00 ± 0.00	13.83 ± 0.18	60.41 ± 1.12
OS_7	24.85 ± 0.21	23.27 ± 0.31	24.37 ± 0.37	1.20 ± 0.10	15.70 ± 0.14	89.39 ± 1.13
OS_8	13.42 ± 0.40	17.76 ± 0.17	15.31 ± 0.92	1.01 ± 0.07	12.83 ± 0.29	60.33 ± 1.85
OS_9	13.11 ± 0.22	15.29 ± 0.42	15.94 ± 0.42	0.54 ± 0.03	12.68 ± 0.50	57.57 ± 1.59
OS_10	23.35 ± 0.09	20.45 ± 0.42	23.37 ± 0.37	0.98 ± 0.08	15.51 ± 0.10	83.65 ± 1.06
OS_11	12.21 ± 0.87	14.03 ± 0.85	20.92 ± 0.43	1.23 ± 0.16	11.17 ± 0.78	59.56 ± 3.09
OS_12	11.77 ± 0.61	14.51 ± 0.18	24.63 ± 0.71	0.48 ± 0.00	12.58 ± 0.20	63.97 ± 1.21
OS_13	13.50 ± 0.33	14.16 ± 0.21	21.13 ± 0.81	0.31 ± 0.04	12.70 ± 0.01	61.80 ± 1.39
OS_14	14.85 ± 0.14	14.17 ± 0.22	17.56 ± 0.58	0.19 ± 0.05	12.62 ± 0.36	59.39 ± 1.35
OS_15	20.93 ± 0.77	23.53 ± 0.66	22.22 ± 0.22	0.11 ± 0.02	15.30 ± 0.52	82.08 ± 2.14
OS_16	16.03 ± 0.22	16.33 ± 0.27	16.88 ± 0.48	0.31 ± 0.04	12.86 ± 0.24	62.41 ± 1.66
OS_17	17.27 ± 0.41	15.09 ± 0.42	15.82 ± 0.23	1.15 ± 0.03	12.65 ± 0.14	61.98 ± 0.42
OS_18	25.31 ± 0.12	25.19 ± 0.56	22.92 ± 0.42	1.10 ± 0.13	15.96 ± 0.23	90.48 ± 1.45
OS_19	21.88 ± 1.01	19.83 ± 0.46	23.93 ± 1.09	0.85 ± 0.05	14.94 ± 0.25	81.42 ± 3.32
OS_20	20.97 ± 0.44	20.60 ± 0.22	21.18 ± 0.58	1.38 ± 0.08	15.04 ± 0.22	79.17 ± 1.54
OS_21	22.22 ± 0.62	19.82 ± 0.27	23.94 ± 0.33	0.97 ± 0.09	14.92 ± 0.25	81.87 ± 1.56
OS_22	22.74 ± 0.33	19.97 ± 0.13	23.76 ± 0.14	0.76 ± 0.13	14.01 ± 0.58	81.25 ± 1.31
OS_23	26.79 ± 0.29	23.53 ± 0.40	24.21 ± 0.63	0.85 ± 0.03	15.81 ± 1.08	91.21 ± 2.43
OS_24	14.40 ± 0.50	14.23 ± 0.15	23.04 ± 0.11	0.58 ± 0.04	11.42 ± 0.09	63.67 ± 0.89
OS_25	15.19 ± 0.13	17.06 ± 0.29	15.10 ± 0.48	1.03 ± 0.06	13.48 ± 0.12	61.86 ± 1.07
Range	11.77–26.79	14.03–25.19	13.66–26.13	0.00–1.59	11.17–18.66	57.57–95.80
PSO	28.61 ± 0.22	25.68 ± 0.24	22.19 ± 0.21	0.50 ± 0.09	14.32 ± 0.17	91.29 ±0.92
SO	0.00 ± 0.00	2.25 ± 0.00	0.00 ± 0.00	10.81 ± 0.01	6.75 ± 0.00	19.81 ± 0.21

## Data Availability

Raw data can be made available upon reasonable request.

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
