# Peer review of "Assessment of Pumpkin Seed Oil Adulteration Supported by Multivariate Analysis: Comparison of GC-MS, Colourimetry and NIR Spectroscopy Data"

_foods, 2022, doi:10.3390/foods11060835_

Round 1

Reviewer 1 Report

Authors aimed to explore the potentialities of sterol composition, NIR data, and colorimetry to assess the authenticity of pumpkin seed oil. However, the meaning of “authenticity” was quite unclear (botanical origin or geographical origin or both?). My major remark concerns the experimental design. Authors planned the laboratory analyses on 25 commercial samples of pumpkin seed oils whose authenticity (whatever it meant) is not reliable (we have to blindly believe in the labelling) and they arbitrarily set a threshold for “authenticity” on the basis of just one analytical parameter (beta-sitosterol percentage). This strongly affected the application of multivariate analyses of experimental data, together with the use of just two “reference” samples as representative of the "variance" of pure pumpkin seed oil and pure “adulterant” (refined sunflower oil).

Moreover, the total sterol content of refined oils widely changes, according to the process conditions, and this could strongly affect the possibility of identifying fraudulent mixing of vegetable oils of different origins on the basis of sterol composition. This aspect was completely disregarded by authors.

In my opinion, the discussion about experimental data is a mix of independent evaluations and often appears just a matter of speculation.

Here are my other comments and suggestions to be taken into consideration:

Page 2, line 52: please add references (EU Regulations that amended the Annex to Regulation (EC) No 510/2006 and added Austrian and Slovenian pumpkin seed oil).

Page 3, lines 101-103: are those samples labelled as PGI? Or just some of them?

Page 3, line 117: maybe “betulin”.

Page 3, lines 148-150: the statement is inconsistent with the formula (1).

Page 4, line 186: what are those limits? (the web page cited in the references was not accessible.

Page 7, lines 231-233: Page 8, lines 242-247: authors should keep to the facts and limit their conjectures. Simply, 12 commercial oils labelled as PSO did not fall within the limits specified in the Croatian disciplinary. Furthermore, in the Materials and Methods section authors did not specify if samples were labelled as PGI. How authors set the threshold for discriminating between authentic and adulterated samples? Authors themselves reported that, according to ref [23], “Δ5-sigmasterol and β-sitosterol increased when 30% of cold-pressed sunflower oil was added to cold-pressed pumpkin seed oil” (Page 5, lines 199-200): how could authors be sure that the other 13 samples were authentic?

Figure 3: what do error bars mean? Authors analysed just one sample of PSO and one of SO.

Figure 4: what kind of data pre-treatment authors applied? (centering, normalization, others?

Reviewer 2 Report

The paper deals with analytical methodologies for the identification of adulterations in pumpkin seed oils.
The authors justify the need for this research on the growing interest that pumpkin seed oil show in the value-added food market, and seek to identify a quick but reliable method to obtain analytical identification of product adulterations.

Introduction

Paper writing and conceptualization appear adequate for the purpose of the journal. Bibliographic sources appear adequate, and the topic is of interest for the scientific community and justifies the purpose of the research.

Materials and methods

It could be questionable the fact that the authors, having chosen to include in the research a number of oils found in retail distribution chains, included as control only a single sample of laboratory prepared pure pumpkin seed oil, albeit according to commercial specifications.
Therefore, authors should justify in the text why a trusted commercial manufacturer was not used to prepare a sample according to market specifications. 

Results and Discussion

The authors found analytical values that prove adulteration in nearly half of the commercial oil samples. This statement is based on the finding of anomalous contents of sterols by GC-MS, attributable to sunflower seed oils. 

However, in the discussion of the results obtained with different methodologies, the classification of adulterated oils by means of multivariate statistical analysis appears confused and needs a thorough rewrite, indicating in greater detail the fundamentals that allowed the categorization of adulterated oils by PLS.

Furthermore, the authors did not include in their manuscript a conclusions paragraph, essential for the description of the final outcome of the paper; also, the authors do not propose concrete application methods for practical implementations.

Reviewer 3 Report

Dear authors,

Reading the manuscript "Assessment of pumpkin seed oil adulteration supported by multivariate analysis: comparison of GC-MS, colorimetry and NIR spectroscopy data" , I realized that the manuscript showed in some parts the scientific rigour wanted, but in other parts I have missed it.The authors have presented critical evaluation in some paragraphs.The references are not exactly current, besides the objective could be more attractive and cientific.Thats why I have written some suggestions in an attempt to improve the paper.

L.18-  the aim in the abstract doesn't seem to me to contemplate all your work, you have done much more than what is here. Don't forget that many researchers initially read only the abstract to verify if the paper is interesting.

L.31- Please cite the authors about " excellent gastronomic and nutritional properties"

L. 46 - the scientific name should come in the 1st time it appears in the paper,

L 89- the authors used the term "fingerprint" in the objectives, is this nomenclature scientific ? I suggest following the same pattern in the Material and methods, results and discussion.

L.93- It was mentioned "possible blends" in the abstract, Introduction and objectives, then throughout the paper they no longer use this nomenclature. Standardize it to ease the reader's understanding. 

L. 94 -98 I suggest rewriting by joining in the previous part L.86-92 and I don't find the part interesting : "In addition, since the standard method for gas chromatographic analysis of the composition of sterols is extremely demanding and expensive", it appears in the paper, being in the objectives.

L.191-  Since you have many figures, figure 1 could be deleted and the result written in the paper. 

L .196- Use also in the objectives and material and methods the nomenclature Δ5-sterols, it only appears in the abstract, results and discussion. I think  Δ5-sterols form is more attractive.

L.274- I usually see L a* b* in tables, I admit that I find it more informative and it would bring more visibility to your results, especially for a*.

L. 398- I could not find the conclusion of the paper. This is very important and I do not know what happened.

L.400- Figure 7 and 8 were not mentioned in the paper.

L 403- I suggest not finishing the paper with a figure

Best regards.

Reviewer 4 Report

The study reported in the manuscript by Balbono et al. aims to determine the sterol content and authenticity of retail oils labelled as pumpkin seed oils and also to investigate the potential of near infrared spectroscopy (NIR) and colorimetry in detecting adulteration.

  1. The scope of this study finds good scientific and practical justification. It fits the scope of the journal. The manuscript is well designed. The references literature is appropriate.
  2. The study is performed systematically and executed up to good standards of analytical spectroscopy. The tools used are well selected and the conclusions drawn are complete in light of the obtained results.
  3. This work has been performed with care and the results are well exposed in the manuscript.

However, this manuscript lacks some clarity and need to be slightly adjusted before publication.

  • There is no summary or conclusion, manuscript seems incomplete in light of both the journal guidelines and the commonly accepted standard for research articles
  • Importantly, while this work is performed with a correct choice of tools and procedures (albeit being established and routine), what stands out is that a dangerously low sample count was gathered for the training set (20) and test set (5)!
  • The study employed a portable NIR spectrometer. It is an important aspect of this work with both very positive implications but certain pitfalls as well. Miniaturized instrumentation features vastly distinct optical configurations / general technical solutions, and very often the performance of the analysis may differ greatly between different instruments on case-to-case basis. It is acceptable that no benchtop spectrometer was used (for example, because of limited availability) for establishing the limit of analytical performance of NIR spectroscopy in this specific application. The manuscript does not consider at all this factor, does not provide the reader with a sufficient awareness about it, suggesting instead that feasibility of “NIR spectroscopy” is examined in general. A brief discussion of the merits and pitfalls of miniaturized instruments would be recommended in the Introduction (for example, on the basis of recent focused reviews such as Eur. J. 2021, 27, 1514-1532)
  • The band assignments / general interpretation of the spectra is much welcomed. However, it is unclear why overtones are put in quotes (i.e. CH “overtone”). Also, NIR spectra are mostly populated by combination bands ( Chem. 2019, 7, 48).
  • In connection with the above, it would perhaps be possible to conclude or assume – why the differences between these oils are expressed in the wavenumber regions visible e.g. in Fig 2? Is it the chemical fingerprint or other effects (instrumental, sample presentation or its physical features) that are expressed therein?
  • The introduction part about used the techniques is quite short, it can be a bit extended. Modest suite of cited references, while this topic is quite popular, could perhaps also be pointed out.
  • As stated in the manuscript, similar study was reported before, where the main difference could appear to be reduced to just different spectral region (perhaps this aspect could be better exposed in the manuscript). Also, the results of the analysis could perhaps be compared with the stated work, e.g. by the statistical metrics.

Round 2

Reviewer 1 Report

I really appreciated the authors' efforts to follow the advice of the reviewers. However, I still think that label as "adulterated" half of the pumpkin seed oil available on the Croatian market could be unfair. I recognize that the unique sterol composition of Cucurbita spp seed lipids would be a useful and valuable tool for assessing the authenticity of pumpkin seed oil, but the huge variability of the sterol composition strongly weaken the conclusions of the paper.
In cold-pressed seed oil from a Tunisian variety (Béjaoui) of Cucurbita maxima, Rezig et al. (Industrial Crops and Products, Volume 37, Issue 1, May 2012, Pages 82-87) reported that sitosterol (39.6%) was the major sterol, followed by Δ5,24-stigmastadienol (21.3%). In cold pressed oil from seeds of Cucurbita pepo of variety “Essahli”, the same authors (Process Safety and Environmental Protection Volume 127, July 2019, Pages 73-81) detected 40% of sitosterol.
According to authors considerations, the above mentioned pumpkin seed oils, should be judged as "adulterated". 

Reviewer 2 Report

The authors gave comprehensive answers to all the questions that have been raised by the reviewer. The conclusions section was also added to the paper, with an adequate summary of the results obtained, although it would be appropriate to include some more references to similar research carried out in the literature. In the opinion of this reviewer, after the revision, the paper has reached a state suitable for publication.

Author Response

The authors gave comprehensive answers to all the questions that have been raised by the reviewer. The conclusions section was also added to the paper, with an adequate summary of the results obtained, although it would be appropriate to include some more references to similar research carried out in the literature. In the opinion of this reviewer, after the revision, the paper has reached a state suitable for publication.

We really thank the Reviewer for his valuable comments and we are glad that he/she acknowledged the revisions made to the manuscript through Revision round 1. Based on his/her comment from Round 2, references were added to conclusion section (L536, 550).